# Stages I–III Inoperable Endometrial Carcinoma: A Retrospective Analysis by the Gynaecological Cancer GEC-ESTRO Working Group of Patients Treated with External Beam Irradiation and 3D-Image Guided Brachytherapy [note 1]

**DOI:** 10.3390/cancers15194750

**Published:** 2023-09-27

**Authors:** Ángeles Rovirosa, Yaowen Zhang, Kari Tanderup, Carlos Ascaso, Cyrus Chargari, Elzbieta Van der Steen-Banasik, Piotr Wojcieszek, Magdalena Stankiewicz, Dina Najjari-Jamal, Peter Hoskin, Kathy Han, Barbara Segedin, Richard Potter, Erik Van Limbergen

**Affiliations:** 1Fonaments Clinics Dpt., Faculty of Medicine, Universitat de Barcelona, 08036 Barcelona, Spain; 2Radiation Oncology Dpt., Hospital Clinic-Universitat de Barcelona, 08036 Barcelona, Spain; 3Cancer Center, Henan Provincial People’s Hospital, Zhengzhou 450003, China; 4Danish Center for Particle Therapy, Institute of Clinical Medicine, Aarhus University, 8200 Aarhus, Denmark; 5Radiation Oncology Dpt., Institut Gustave Roussy, 94805 Villejuif, France; 6Radiation Oncology Dpt., Radiotherapie Groep, 6815 AD Arnhem, The Netherlands; 7Maria Sklodowska-Curie National Research Institute of Oncology, 44-102 Gliwice, Poland; 8Institut Català d’Oncologia, l’Hospitalet de Llobregat, 08908 Barcelona, Spain; 9Mount Vernon Cancer Centre, Northwood, Middlesex HA6 2RN, UK; 10Radiation Oncology Dpt., Princess Margaret Hospital, University Health Network, Toronto, ON M5G 2M9, Canada; 11Radiation Oncology Department, Institute of Oncology Ljubljana, Faculty of Medicine, University of Ljubljana, 1000 Ljubljana, Slovenia; 12Department of Radiation Oncology, Medical University of Vienna, 1090 Vienna, Austria; 13Radiation Oncology Dpt., University Hospital Gasthuisberg, 3000 Leuven, Belgium

**Keywords:** inoperable endometrial carcinoma, 3D-image-guided-brachytherapy

## Abstract

**Simple Summary:**

About 10% of early-stage endometrial cancers (EC) are inoperable. Curative treatment is still recommended in these cases. The aim of this retrospective study, which is the largest in the literature, was to analyse the outcomes of stages I–III EC patients treated with the intention to cure by external beam irradiation (EBRT) and 3D-image-guided brachytherapy (IGBT). EBRT+IGBT provides good cancer-specific survival results of 88.7% and 71.2% at 2 and 5 years, respectively. A few patients without uterine relapses developed nodal or systemic relapses, highlighting the importance of uterine control.

**Abstract:**

Background/Purpose: Analyse the outcomes of stages I–III inoperable endometrial cancer (IEC) patients treated with external-beam-irradiation (EBRT) and 3D-image-guided-brachytherapy (IGBT). Material and Methods: Medical records of IEC patients receiving EBRT + IGBT in eight European and one Canadian centres (2004–2019) were examined, including: pelvic ± para-aortic EBRT and lymph node boost; anaesthetic procedure, applicators, BT-planning imaging, clinical target volume (CTV), brachytherapy schedule, and EQD2 to the CTV_(α/β=4.5Gy)_ and D2 cm^3^_(α/β=3Gy)_ for organs at risk. Complications are evaluated using CTCAEv4 scores. The 2- and 5-year survival probability according to stages was estimated (cancer-specific survival (CSS), disease-free survival (DFS), local relapse-free survival (LRFS), loco-regional relapse-free survival (LRRFS), and distant metastasis-free survival (DMFS)). Statistics: descriptive analysis and the Kaplan–Meier method. Results: 103 patients (stages: I-44, II-14, III-44) were included. Median follow-up: 28 months (7–170). All patients received pelvic ± para-aortic EBRT. Median D90-EQD2_(α/β=4.5)_ to the CTV:73.3 Gy (44.6–132.7), 69.9 Gy (44.7–87.9 and 75.2 Gy (55.1–97) in stages I, II, and III, respectively. Thirty patients presented relapse (stages: 10-I, 3-II, 17-III): 24 uterine (stages: 7-I, 3-II, 14-III), 15 nodal (stages: 4-I, 1-II, 10-III), and 23 distant (stages: 6-I, 2-II, 15-III). Five year CSS was 71.2% (stages: 82%-I-II and 56%-III) and DFS, LRFS, LRRFS, and DMFS were 55.5%, 59%, 72%, and 67.2%, respectively. Late G3-G4 complications (crude): 1.3% small bowel, 2.5% rectum, and 5% bladder. Conclusion: In stages I–III of the IEC, EBRT + IGBT offer good 2- and 5-year CSS of 88.7% and 71.2%, respectively, with the best outcomes in stages I–II. Prospective studies are needed to determine how better outcomes can be achieved.

## 1. Introduction

Endometrial cancer (EC) is the most common gynaecological cancer, and the incidence is expected to rise in developed countries in the forthcoming years because of longer life expectancy. In the United States, 51,065 cases were diagnosed in 2020, with an expected increase up to 125,922 new cases in 2040 [1]. The main treatment is surgery ± radiotherapy ± chemotherapy, but about 10% of early-stage ECs are inoperable due to the body mass index (BMI), age-associated diseases, or anaesthetic contraindications [2,3,4]. Palliative approaches, such as hormone treatment, external beam irradiation (EBRT), and sometimes chemotherapy, have been described in the literature [3]. Nevertheless, curative treatment of these inoperable patients is considered in the European Society of Gynaecological Oncology (ESGO-ESTRO-ESP) and Federation International of Gynaecological Oncology (FIGO) 2021 recommendations [5,6].

Most series have performed treatment using 2D-planning for brachytherapy (BT), although a few have begun using 3D-image-guided BT (IGBT), mainly in stage I [7]. However, to avoid delays in surgery, 3D-IGBT was recommended during the first part of the COVID-19 pandemic [8].

In a recent retrospective study by the *Groupe Européen* de Curiethérapie (GEC) and the *European* Society for Radiotherapy & Oncology (ESTRO) Gynae-Working Group, 62 stage I patients treated with 3D-IGBT alone were analysed, with 2- and 5-year results showing a cancer-specific survival (CSS) of 93.3 and 80.5%, respectively, a disease-free survival (DFS) of 84.8% and 80.5%, respectively, local relapse-free survival (LRFS) of 93.1% and 88.7%, respectively, locoregional relapse-free survival (LRRFS) of 91% and 91%, respectively, and distant metastases-free survival (DMFS) of 90.2% and 90.2%, respectively. CSS at 2 and 5 years was better in stage IA vs. IB, with 96.6% and 88.7%, respectively, vs. 77.5% and 64.6%, respectively (*p* = 0.043). Late G3 complication rates of 2.1% appeared in the vagina and bladder, respectively [7]. These results were better regarding complications compared with other previous cohorts treated with radiotherapy [4,9]. Although BT alone can be performed in some stage I patients (stages IA and G1–G3 and IB and G1–G2), stages I–III can be treated with EBRT + 3D-IGBT with curative aim, as reported in the last European Society of Gynaecological Oncology (ESGO), ESTRO, and the European Society of Pathology (ESGO-ESTRO-ESP) and American Brachytherapy Society guidelines. EBRT + IGBT is considered for high-grade IEC tumours and/or deep myometrial invasion and in advanced stages [5,9].

The aim of the present study was to analyse the characteristics and outcomes of 103 stage I–III patients treated with curative aim with EBRT + 3D-IGBT, which is the largest series in the literature. We also propose recommendations for treatment according to the results.

## 2. Material and Methods

This study received Institutional review board approval, and all the patients provided signed informed consent (HCB/2018/0341).

The medical records of 225 patients were available, and finally, the data from 103 inoperable EC patients treated with EBRT + 3D-IGBT between 2004 and 2019 were analysed. These patients were from eight European and one Canadian centre participating in the Gynae GEC-ESTRO-Working Group. All patients had pathologically confirmed EC, and the FIGO 2009 classification was used for staging and diagnostic imaging prior to treatment [10]. The most common contraindications for surgery were a BMI > 35, anaesthetic contraindications, advanced age, and different comorbidities impeding surgical treatment. The inclusion criteria included the availability of pelvic computerised tomography (CT), magnetic resonance imaging (MRI), or positron emission tomography (PET)-CT, 3D-IGBT, and a minimum 6-month follow-up to evaluate the uterine tumour and nodal response.

The following data were recorded: age, FIGO 2009 stage, tumour grade, pathological subtype, largest tumour diameter in mm, diagnostic imaging method, EBRT data, 3D-imaging prior to BT, anaesthetic procedure (none, local, spinal, and general), applicator type, BT-planning imaging, gross tumour volume (GTV), clinical target volume (CTV) considered, high-dose-rate (HDR) vs. pulsed-dose-rate (PDR) treatment, BT schedule, total EBRT + BT dose in EQD2 (total dose equivalent to 2 Gy fractions), using an α/β = 4.5 Gy as repair half time for the CTV and GTV (sum of EBRT+BT doses), and in the case of organs at risk (OAR) an α/β=3 in D2 cm^3^. Complications were evaluated using CTCAEv4 scores [11].

Statistics: The 2- and 5-year survival probability was estimated for CSS, DFS, LRFS (uterus), LRRFS (relapse in pelvic and/or para-aortic lymph nodes), and DMFS. Survival analysis included descriptive analysis and the Kaplan–Meier method. The impact of prognostic factors on CSS and LRFS was assessed by log-rank and Breslow tests [12,13].

## 3. Results

This retrospective multicentre study included 21 patients from The Institut Gustave Roussy (France), 15 from the Radiotherapiegroep Arnhem, Arnhem (The Netherlands), 13 from the Maria Sklodowska-Curie National Research Institute of Oncology in Gliwice, Gliwice (Poland), 13 patients from the Institut Català d’Oncologia, Hospitalet de Llobregat (Spain), 12 patients from the Mount Vernon Cancer Centre (United Kingdom), 9 patients from UZ Leuven (Belgium), 7 patients from the Hospital Clinic-Universitat de Barcelona, Barcelona (Spain), 7 patients from The Princess Margaret Hospital, Toronto (Canada), and 6 patients from the Institute of Oncology of Ljubljana, Ljubljana (Slovenia).

### 3.1. Patients and Tumour Characteristics

The median age of the patients was 71 years (range 45–106), and the median follow-up was 28 months (range 7–170). Staging was as follows: 44 patients were in stage I (14-IA (13.6%)), 30 in IB (29.1%)), 15 in stage II (14.6%), 44 in stage III (5-IIIA (4.9%)), 14 in IIIB (13.7%), 11 in IIIC1 (10.7%), and 14 in IIIC2 (13.6%).

Table 1 shows the pathologic characteristics of the patients by stage. The most common pathological type was endometrioid, and the most common tumour grade was grade 2, followed by grades 1 and 3.

### 3.2. Treatment

#### 3.2.1. External Beam Irradiation

Table 2 shows the planning and EBRT schedules and the doses to the pelvic and para-aortic areas by stage.

Most treatments were performed after 3D planning, followed by intensity-modulated radiotherapy/volumetric modulated arc radiotherapy (IMRT/VMAT), with the latter also being used for nodal boost.

Pelvic nodal boost was performed in 11 patients with pelvic node involvement with a dose ranging from 48.9–66 Gy.

Prophylactic para-aortic irradiation was performed in 25 stage IA-IIIC2 patients, depending on the centre criteria, with doses ranging between 29.7 and 50.4 Gy EQD2_(α/β=4.5Gy)_. Nevertheless, 4/14 patients in stage IIIC2 did not receive para-aortic irradiation, and 5/14 in stage IIIC2 received a para-aortic node boost to a total dose ranging from 55–63.7 Gy.

#### 3.2.2. 3D-Image-Guided Brachytherapy

1. Here, 3D-IGBT prior to BT was obtained in 73/103 patients who had undergone >1 imaging modality prior to BT: CT in 78/103 patients, ultrasound (US) in 59/103 patients (in two patients, this information was missing), MRI in 62/103 patients (this information was missing in three patients), and PET or PET-CT was performed in 34/103 patients (this information was missing in 4).

2. The most common anaesthetic procedure was general anaesthesia in 44/103 patients (42.7%) and spinal anaesthesia in 18/103 patients (17.5%). Local cervical anaesthesia and no anaesthesia were less frequent in 15/103 (14.6%) and 13/103 patients (12.6%), respectively). This information was not available in 13/103 (12.6%) cases.

3. The most common definition of the CTV was the uterus and cervix in 28% of the patients, followed by the uterus + cervix + upper 1/3 of the vagina in 25.3%, the uterus + cervix + parametria in 18.4%, the uterus in 13.6%, and the uterus + cervix + upper 1/3 of the vagina + parametria in 1.9%. This information was missing in 3.9%. GTV was delineated in 89/103 patients, and information on doses was only available in 23/103 patients.

4. Imaging procedures used for IGBT planning were US in 37/103 (35.9%) patients, CT in 77/103 (74.5%) patients, and MRI in 33/103 (32%) patients.

5. *Applicators*. Intrauterine tandem (IUT) was the most common applicator in 61/103 (D59.2%) patients, being exclusively used in 16/103 (15.5%) patients, with a vaginal mould in 21/103 (20.4%), and associated with colpostats in 16/103 (15.3%) and with vaginal cylinders in 6/103 (5.8%) patients (in two patients interstitial treatment was added) and in 2/103 (1.8%) combined with interstitial treatment (this treatment can be contraindicated in patients receiving anticoagulant treatment). A Y-Shaped-Rotte was used in 25/103 patients (24.3%), Norman Simon in 16/103 (15.5%), and finally, one (0.9%) patient was treated by IUT and also Norman Simon in two different insertions.

6. *Doses per fractionation, number of insertions, and specification.* The dose specification was: 1 to 4 cm from the IUT/Y-Shaped Rotte applicator in 41/103 (39.8%) patients, to the inverse point A in 22/103 patients (21.4%), to the CTV in 20/103 (19.4%), to serosa in 12/103 (11.7%) patients, and to the serosa and 2 cm from the applicator in two different insertions in 3/103 (2.9%) patients. This information was missing in 5/103 (4.9%) patients.

HDR was used in 62/103 (60.2%), PDR in 40/103 (38.8%), and both were used in one patient (0.9%). In HDR treatments, the median dose per fraction was 7 Gy, ranging from 4–8.5 Gy. The median number of fractions was four (range between 1 and 7) with a median of three insertions (range 1–6). Information related to the dose per fraction and the number of insertions was missing in 25 and 13 cases, respectively. In PDR treatments, the median pulse size was 0.5 Gy/hr (range 0.25–30), but in three patients, this information was not reported. The median number of pulses was 41.5, ranging between 1 and 30 and 2 in 6 patients.

7. *Doses to the CTV and GTV.* Overall, D90 median doses and ranges for the CTV and GTV EQD2_(α/β=4.5Gy)_ were 75 Gy (55–132.7) and 85.4 Gy and 80, depending on the case (missing in three patients). The number of insertions was 1 in 34 patients (58.5–189.7). Table 3 shows the D90 in CTV and GTV by stages.

## 4. Outcomes

With the present follow-up, 22/103 patients died of EC, 21/103 died of non-cancer-related diseases, and in 9/103 patients, the cause of death was not reported.

### 4.1. Relapses

*Uterine (local) relapses*. There were 24 (18 uterine relapses and 6 cases of uterine disease persistence) after treatment with the diagnosis based on imaging, and this information was missing in two patients. Ten of these 24 patients presented with isolated uterine relapse/persistence, and 8/10 underwent rescue surgery, 6 of whom were cured after this treatment.

Table 3 shows that uterine relapse or persistence was presented by 7/44 (15.9%) stage I patients, 3/15 (20%) stage II patients, and 14/44 (31.8%) patients in stage III.

In 15/24 cases with uterine relapse or persistence, the D90 CTV dose was known, and 6/15 (40%) patients received D90 ≤ 70 Gy EQD2_(α/β=4.5Gy)_ (range 59 Gy to 70 Gy) and 9/15 (60%) > 70 Gy (range 73–97 Gy).

Doses to the GTV were reported in 23/103 patients, and in these there were four uterine relapses; of these 23 patients, 2 of 13 (15.4%) cases with GTV D90 EQD2_(α/β=4.5Gy)_ ≤ 85 Gy (66.1 and 76 Gy) had a relapse, and 2/10 (20%) cases with GTV D90 > 85 Gy (97.5 and 121.3 Gy) had a local relapse.

Lymph node relapse developed in the pelvis in 15 patients (4744 (9.1%)) in stage I, 2/15 (13.3%) in stage II, and 9/44 (20.5%) in stage III. Para-aortic lymph node relapse was present in 12/103 patients, and data were missing in 1/103 patients (2/44 (4.5%)) in stage I, 1/15 (6.6%) in stage II and 9/43 (20.9%) in stage III. Pelvic and/or para-aortic and/or distant metastases developed in 18/24 (85%) patients with uterine tumour progression. Only four patients had pelvic and/or para-aortic and/or distant metastasis without uterine relapse (1-IA, 1-IIIC1, and 2-IIIC2).

Distant metastasis appeared in 24 patients and was common in stage III (15/44). In 7/24 patients, metastasis was without local recurrence (2-IA, 1-IB, 1-II, 1-IIIB, and 2-IIIC2), and in eight patients, this information was missing. Three patients had distant metastasis without local or nodal relapse (2-IA and 1-IIIB).

### 4.2. Prognostic Factors

On univariate analysis, only stage had an impact on CSS, but not age, BMI, pathological type, grade, uterine length, applicator type, or topography in the fundus. A D90 dose to CTV > 75 Gy vs. ≤75 Gy also had no impact on CSS.

### 4.3. Survival Analysis

There was no difference in CSS, LRFS, LRRFS, and DMFS among patients with stages I and II, and therefore the survival analysis was performed considering stages I–II vs. III. Figure 1 shows the different results in survival at 2 and 5 years, and Figure 2 depicts the results considering stages I + II vs. III. The impact on DFS was mainly related to uterine and node relapses.

### 4.4. Late Complications

Late bladder complications were not reported in 23/103 patients and were graded as G0 in 56/80 (70%), G1 in 15/80 (18.8%), G2 in 3/80 (6.3%), G3 in 3/80 (3.7%), and G4 in 1/80 (1.2%). Late rectum complications were not available in 24/103 patients but were considered G0 in 68/79 (86.1%), G1 in 7/79 (8.9%), G2 in 2/79 (2.5%), and G3 in 2/79 (2.5%). Late sigmoid complications were reported in 68/103 patients, with G1 complications in 5/68 (7.3%) and G2 toxicity in 1/68 (1.5%), while no complications were reported in 62/68 (91.2%). Late small bowel complications were not reported in 24/103 patients; 73/79 (92.4%) patients did not develop any complications; 4/79 (5.1%) were considered as G1, 1/79 (1.3%) as G3, and 1/79 (1.3%) as G4. Late vaginal complications were missing in 26/103 patients, and in 69/77 (89.6%) vaginal complications were absent, being present as G1 in 7/77 (9.1%) and as G2 in 1/77 (1.3%).

## 5. Discussion

The aim of the present study was to analyse the characteristics and outcomes of 103 stage I–III patients treated with curative EBRT + 3D-IGBT. The studies evaluating 3D-IGBT were retrospective and had a small sample size [4,14,15,16,17,18,19,20,21]. Two studies by the National Cancer Institute and the Surveillance, Epidemiology and End Results programme, demonstrated that BT ± EBRT has the best results in IEC [4,22]. In a case series and systematic review of the literature, it was concluded that definitive IGBT ± EBRT provides good local control with low toxicity for IEC [23]. The present retrospective series is, to date, the largest in the literature of patients treated with 3D-EBRT + IGBT and there is a lack of prospective trials. Table 4 shows the different series in the literature using EBRT + IGBT in MRI planning [4,14,15,16,17,18,19,20].

The 2015 ABS consensus on IEC recommended exclusive BT in patients with stage IA, G1-2, and performing an MRI to analyse the uterus and possible pelvic disease, while EBRT+BT was recommended in the remaining patients [9]. IEC patients receiving EBRT+ BT should be well staged using imaging techniques, such as MRI and PET-CT, to determine the possible presence of myometrial invasion and tumour extension, as well as lymph node and distant metastasis status. The US can also help determine the depth of uterine infiltration [24]. In the present series, all the patients were staged to adapt the treatment to each case.

A previous study by the Gyn GEC-ESTRO Working Group in stage I patients treated with exclusive BT showed similar results to those of the series that used MRI for BT planning with GTV definition [7]. In the present series, the 5-year CSS was 82% in stages I and II, with no differences between these two stages. Nevertheless, prognostic factors such as G2-3, myometrial invasion ≥50% and large tumour size were more common in the present series compared to the series by our group in patients treated with exclusive BT [7]. In view of these worse prognostic factors, in the present series, EBRT was necessary to provide similar results in stages I–II patients. The 5-year CSS in stage III patients was somewhat similar to that of stage III postoperative cases treated with EBRT + BT [25]. Thus, EBRT + BT offered great survival benefits to patients not suitable for surgery. These survival results are similar to those reported in the small number of 3D-based series shown in Table 4. Moreover, we can hypothesise that these results could improve with an increased dose and the use of chemotherapy, which is indicated in these patients considering their general status as shown in the PORTEC-3 trial [25].

General anaesthesia was most commonly used in IEC patients, with the type of anaesthesia used apparently being related to the centre or patient characteristics. In this series, CTV delineation was not homogeneous, but the results showed that the uterus and cervix were included as CTV in almost all the cases. The ABS recommends the inclusion of the upper third of the vagina in the CTV. The upper third of the vagina was included in 27% of the present cases. Although the lack of vaginal relapses in the present series suggests that the vagina need not be included in the CTV, the retrospective nature of the analysis does not allow us to establish this as a recommendation. The different CTV and GTV definitions worldwide (Table 4) need common guidelines and prospective studies to confirm whether the vagina should or should not be included as CTV and which CTV and GTV are best in each case and stage of the IEC.

Of particular interest is that patients with uterine relapse developed pelvic, para-aortic, and distant metastasis. On the other hand, the reduction in lymph nodes and distant metastasis was noteworthy, highlighting the importance of achieving uterine control.

In the present series, patients received similar D90 CTV EQD2_(α/β=4.5Gy)_ in the different stages. The GTV EQD2_(α/β=4.5Gy)_ was only obtained in 23 patients, with the lowest dose values for stage III. Nevertheless, although we could not establish a relationship between the dose received in the GTV and CTV and uterine control, the results obtained in the present series are good.

Para-aortic irradiation was administered to 25 patients in the entire series, but only 5/14 IIIC2 received a lymph node boost, and 9% of para-aortic node relapses were present in stage III patients. Age and comorbidities may have influenced the lack of para-aortic boost in IIIC2 patients.

The main limitation of the present study is its retrospective nature; the heterogeneity in CTV definition and MRI for planning was only used in 32% of cases (24% used the Y-shaped Rotte technique). Despite this, EBRT+IGBT provided very good results with few late complications and should be used whenever possible in IEC patients.

## 6. Conclusions

In conclusion, EBRT + IGBT is a good curative option in inoperable endometrial cancer stages I–III, offering a 2- and 5-year CSS of 88.7% and 71.2%, respectively. In the present cohort, local uterine control seems to be of great importance since so few patients without uterine relapses or persistence presented nodal or systemic relapses. Moreover, despite the risk in these patients, salvage hysterectomy should be considered whenever possible, while always taking into account the clinical status of the patients. Prospective studies are needed to establish the most adequate relationship between the doses administered and the CSS. Other ways to improve local control include the development of specific guidelines for volume definition and prospective studies of brachytherapy doses. Additionally, radiotherapy to pelvic and para-aortic nodes should be recommended in the guidelines, and chemotherapy in stage III and serous types may be beneficial as recommended by PORTEC-3.

## Figures and Tables

**Figure 1 cancers-15-04750-f001:**
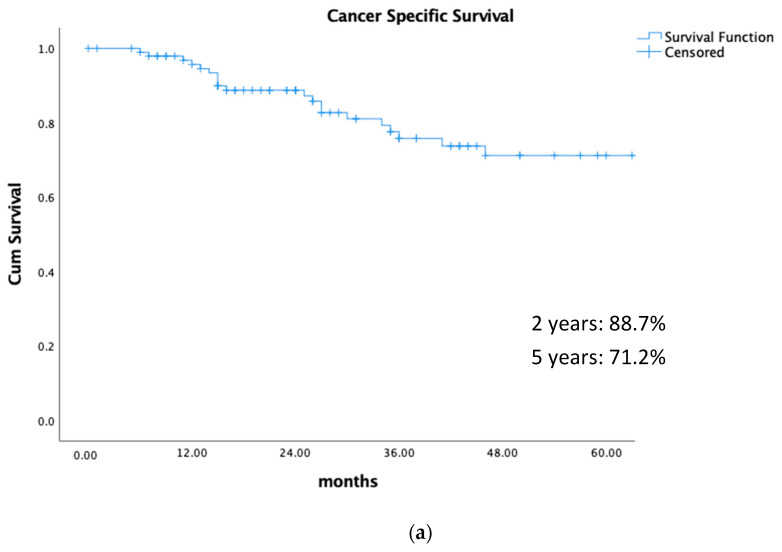
Different survivals at 2 and 5 years for the entire series. (**a**) Cancer-specific survival. (**b**) Disease-free survival. (**c**) Local (uterine) relapse-free survival. (**d**) Lymph node relapse (LRR)-free survival. (**e**) Distant metastasis-free survival.

**Figure 2 cancers-15-04750-f002:**
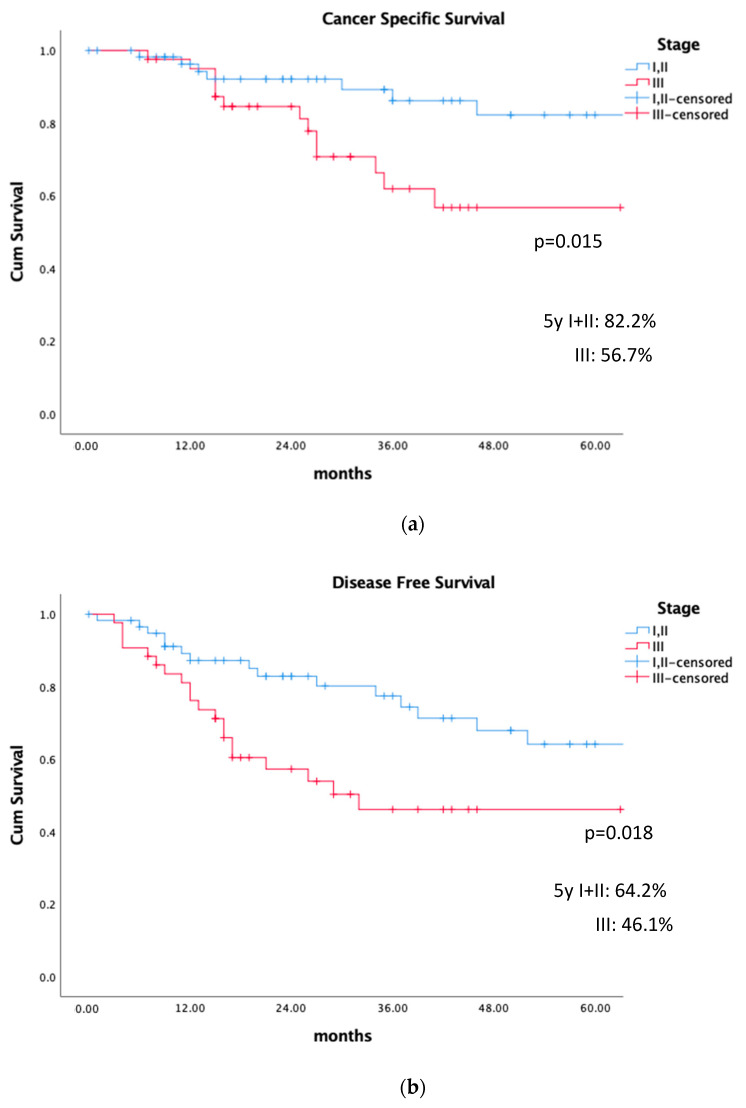
Survival rates in stages I–II vs. III at 5 years. (**a**) Cancer-specific survival. (**b**) Disease-free survival. (**c**) Local-relapse (uterus)-free survival. (**d**) Lymph node relapse-free survival (LRRFS). (**e**) Distant metastasis-free survival.

**Table 1 cancers-15-04750-t001:** Pathologic characteristics of the patients by stages.

	All Stages (103)	STAGE I	STAGE II	STAGE III	Missing Information
		(44)	(15)	(44)	
BMI ≥ 35	39/77 (50.6%)	21/33 (63.6%)	9/10 (90%)	9/34 (26.55)	26 (25.3%)
BMI < 35	38/77 (49.4%)	12/33 (36.4%)	1/10 (10%)	25/34 (73.5%)	
Pathological Type:					4 (3.8%)
Endometrioid	82 (79.6%)	33 (75%)	15 (100%)	34 (77.3%)	
Serous,	4 (3.9%)	1 (2.3%)	0 (0%)	3 (6.8%)	
Clear cell	2 (1.9%)	2 (4.5%)	0 (0%)	0 (0%)	
Mixed	5 (4.9%)	3 (6.8%)	0 (0%)	2 (4.5%)	
Other (no carcinosarcoma)	6 (5.8%)	1 (2.3%)	0 (0%)	5 (11.4%)	
Grade:					18 (16.4%)
1	30 (29.1%)	16 (36.4%)	5 (33.3%)	9 (20.5%)	
2	36 (81.8%)	14 (31.8%)	5 (33.3%)	17 (38.6%)	
3	20 (19.4%)	5 (11.4%)	3 (20%)	12 (27.3%)	
Myometrial invasion:					44 (42.8%)
<50%	28 (27.2%)	15 (34.1%)	5 (33.3%)	8 (18.2%)	
≥50%	53 (51.5%)	14 (31.8%)	8 (53.3%)	31 (70.5%)	
Median tumour size (mm) and range.	55.5 (13–100)	47	82.5	60	57 (55.3%)
		(13–80)	(37–100)	(16–98)	
Tumour site in uterus:					27 (26.2%)
Fundus	39 (37.9%)	10 (22.7%)	9 (60%)	20 (45.5%)	
The rest of tumour sites	37 (36.0%)	16 (36.4%)	4 (26.7%)	17 (38.6%)	

BMI: body mass index.

**Table 2 cancers-15-04750-t002:** External beam irradiation.

		Missings
EBRT Technique:		8
2D	6	
3D CRT	60	
VMAT/IMRT	47	
Image for Planning:		
CT with contrast	17	16
CT without contrast	88	11
Image fusion		
MRI	19	9
PET CT	16	9
No image fusion	68	9
Doses (Gy) (median and ranges):		
Pelvic elective target EQD2 *	46 (29.7–60)	
Para-aortic elective target EQD2 *	44 (29.7–50.6)	
		1
Pelvic Node boost	11	15
Seq	4	
SIB	56.7 (48.8–66)	0
Pelvic nodal EQD2 *		
		0
Para-aortic Node Boost dose	5	
Seq	2	
SIB	3	1
Para-aortic nodal EQD2 *	63.6 (55.3–63.7)	

* EQD2: α/β = 4.5. Seq: sequential; SIB: simultaneous integrated boost; CT: computerized tomography; MRI: magnetic resonance imaging; PET: positron emission tomography; CRT: conformal radiotherapy; IMRT: intensity-modulated radiotherapy; VMAT: volumetric modulated arc radiotherapy.

**Table 3 cancers-15-04750-t003:** Doses to the clinical target volume, gross tumour volume at the time of brachytherapy and relapses.

	All Stages (103)	Stage I (44)	Stage II (15)	Stage III (44)	Missing Data
Reported CTV doses:					
reported D90 dose					
Median BT dose (Gy) EQD2_(α/β=4.5)_		31	5	14	53
		26.1 (8.6–86.7)	22.6 (2.8–41.9)	30.4 (11.5–48.1)	
Median EBRT + BT D90 (Gy) EQD2_(α/β=4.5)_	27.4 (2.8–86.7)				
	73.3 (15–132.7)	73.3 (44.6–132.7)	69.9 (15–87.9)	75.2 (55.1–97)	34
reported GTV doses					
reported D90 dose					
Median BT dose (Gy) EQD2_(α/β=4.5)_		5	5	13	80
	40.1 (3–146)	46.1 (15–146)	39.5 (3–77)	33.9 (16–91)	
Median EBRT + BT D90 (Gy) EQD2_(α/β=4.5)_	86.1 (49–190)	89.7 (59–190)	87.8 (49–121)	77.5 (63–135)	80
Uterine Relapse/Persistence	24/101	7	3	14	2
Pelvic Nodal Relapse	15/101	4	2	9	2
Para-aortic Node Relapse	12/99	2	1	9	4
Distant Metastases	24/95	7	2	15	8

CTV: Clinical Target Volume; GTV: Gros tumoral Volume; EBRT: External beam irradiation; BT: Brachytherapy.

**Table 4 cancers-15-04750-t004:** Retrospective studies in the literature using EBRT + IGBT.

Author	Number of Patients	Stage	CTV	Median D90 GTV GTV (Gy) QD2_(α/β=10)_	Median D90 CTV CTV (Gy) EQD2_(α/β=10)_	Relapses	OS% (y)	CSS% (y)	DFS% (y)
Gill et al., 2014 [14]	14	I	Uterus + cervix + 1–2 cm vagina	138.0 ± 64.6 (mean ± SD)	72.4 ± 6.0 (mean ± SD)	1 local relapse	2 y: 94.4	-	-
Archaya et al., 2016 [15]	15	I–III	uterus and cervix	-	EBRT: 48–50.4	2/15 pelvis 5/15 M1	2 y: 64	-	-
					BT: 37 (31–56)				
Jordan et al., 2017 [16]	9	I	GTV + 2 cm	100.5 (70.5–177.1)	60.9 (48.1–74)	1 local relapse 93.4% 4 y	-	-	-
Gebhardt et al., 2019 [17]	16	I	Uterus + cervix + 1 cm vagina	115 (101.2–131.0)	95.1 (52.8–116.2)	96.8% 2 y	2 y: 96.8	-	90.1 2 y
Espenel et al., 2020 [18]	27	I–IV	Uterus + cervix + outer residual tumour	73.6 (64.1–83.7)	60.7 (56.4–64.2)	-	5 y: 63	-	49.7 5 y
Gannavarapu et al., 2020 [4]	25	I–III	Uterus, cervix, + upper third of vagina	-	70.2 (51–88.7)	-	2 y: 75	2 y: 100	
Carpenter et al., 2023 [20]	32	I–IV	HR-CTV	-	HR-CTV > 91.3	Favourable 100%.	2 y: 65	-	77 2 y
			IR-CTV		(88.8–102)	Unfavourable 50%			
					IR-CTV > 73.4 (69.9–76)				
Huang et al., 2023 [21]	50	I–III	Uterus + cervix + 1 cm vagina	166.2 (123–189.8)	72.9 (74.9–80.3)	-	2 y: 75	2 y: 83	-
Present series	103	I–III	Most uterus + cervix	23/103	53/103	5 y: 18 uterine	-	5 y:	5 y:
				85.4 (58.5–189.79)	75 Gy (55–132.7)	27 nodal		I–II: 82.2	I–I:64.2
				(α/β = 4.5)	(α/β = 4.5)	24 distant		III 56.7	III:46.1

CTV: Clinical target volume. GTV: Gross tumoural volume. Y: years; HR-CTV: endometrium + cavity + tumour mass. IR-CTV: HR-CTV + 5 mm excluding organs at risk.

## Data Availability

Data is unavailable due to privacy or ethical restrictions.

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
