# Peer review of "Stages I–III Inoperable Endometrial Carcinoma: A Retrospective Analysis by the Gynaecological Cancer GEC-ESTRO Working Group of Patients Treated with External Beam Irradiation and 3D-Image Guided Brachytherapy"

_cancers, 2023, doi:10.3390/cancers15194750_

Round 1

Reviewer 1 Report

First of all i would like to thank authors for submitting such an impressive work wich is the largest in the literature of patients treated with 3D- EBRT+IGBT and there is a lack of prospective trials.

 I must admit it is complete and well described .

A few typographical omissions need to be corrected: 

Abstract Ligne 53: "and67.2% respectively.

Table 3 : "GTVreported doses"

Author Response

Thank you so much for your comments, and also thank you for your sugestions  that will be corrected.

Reviewer 2 Report

Please find a few textual comments in the uploaded paper. In my opinion this is an important paper with practice-changing potential. 

I was, however, interested in the question whether type of applicator ( Rotte vs tandem) had any influence on the uterine control rate. I would surmise that an Y-shaped  applicator will produce better coverage of the whole endometrium and therefore better local control. I would like more attention  for this issue in the results and discussion  sections. 

Author Response

I have corrected all your comments. IN the case of Y-Shape Rotte vs tandem in te present series as hapens in others there is not impact in local control. I do not know the reason. The last has not been included in the text. 

Reviewer 3 Report

1.       The efficacy profile of brachytherapy in patients with endometrial cancer not eligible for surgery.

2.       The topic is relevant and information still lacks in this domain.

3.       Contributes to the development of this procedure and allows pondering further questions worth investigating.

4.       The methodology allows an accurate depiction of expected outcomes.

5.       The conclusion supports the hypotheses

6.       References are adequate for the manuscript

7.       No additional comments.

Before acceptance, English editing should be extensively performed throughout the manuscript. I suggest employing professional services to this end.

Author Response

Thank you so much for your comments, English will be edited

Round 2

Reviewer 3 Report

Editings have enhanced readability and ideas will be much easily transmitted